# Microbiome-Related and Infection Control Approaches to Primary and Secondary Prevention of *Clostridioides difficile* Infections

**DOI:** 10.3390/microorganisms11061534

**Published:** 2023-06-09

**Authors:** Lynne V. McFarland, Ellie J. C. Goldstein, Ravina Kullar

**Affiliations:** 1McFarland Consulting, Seattle, WA 98115, USA; 2R.M. Alden Research Laboratory, Santa Monica, CA 90404, USA; ejcgmd@aol.com; 3Expert Stewardship Inc., Newport Beach, CA 92663, USA; ravina.kullar@gmail.com

**Keywords:** *C. difficile* infections, prevention, live biological product, probiotics

## Abstract

*Clostridioides difficile* infections (CDIs) have decreased in the past years, but since 2021, some hospitals have reported an increase in CDI rates. CDI remains a global concern and has been identified as an urgent threat to healthcare. Although multiple treatment options are available, prevention strategies are more limited. As CDI is an opportunistic infection that arises after the normally protective microbiome has been disrupted, preventive measures aimed at restoring the microbiome have been tested. Our aim is to update the present knowledge on these various preventive strategies published in the past five years (2018–2023) to guide clinicians and healthcare systems on how to best prevent CDI. A literature search was conducted using databases (PubMed, Google Scholar, and clinicaltrials.gov) for phase 2–3 clinical trials for the primary or secondary prevention of CDI and microbiome and probiotics. As the main factor for *Clostridium difficile* infections is the disruption of the normally protective intestinal microbiome, strategies aimed at restoring the microbiome seem most rational. Some strains of probiotics, the use of fecal microbial therapy, and live biotherapeutic products offer promise to fill this niche; although, more large randomized controlled trials are needed that document the shifts in the microbiome population.

## 1. Introduction

*Clostridioides difficile* infections (CDIs) are caused by a spore-forming Gram-positive anaerobic bacterium and may result in a spectrum of disease states, ranging from asymptomatic carriage to mild–moderate diarrhea to colitis or pseudomembranous colitis. The number of annual cases of CDI in the USA averages over 500,000/year with over 29,000 associated deaths, while in Europe, the yearly average is over 150,000 cases with 8300 deaths/year [1,2]. The Centers for Communicable Diseases identified CDI as an ‘urgent health threat’ in 2020 and reported that CDI is a leading cause of healthcare-associated gastrointestinal infections [1]. The annual incidence of CDI continually increased from the 1980s to 2000, when the rate peaked during outbreaks of the hypervirulent BI/NAP-1/027 strain (reaching 121.2/100,000 person-years in the US) but since then, the rate has slowly decreased due to enhanced infection control practices and a better understanding of how this pathogen is transmitted [1,3]. CDI rates during the COVID pandemic have decreased in some US and European hospitals, yet have increased or remained stable at other healthcare facilities [4]. The consequences of CDI include increased hospital stays (5–10 days), increased intensive care unit admissions (5–18%), increased attributable mortality (2–7%), increased healthcare costs (USD 71,980 to over USD 200,000/patient), requiring a colectomy (1–9%), and development of recurrent CDI (20–40%), which carries an increased risk of sepsis (17–43%) [3,5,6,7,8,9,10]; CDI remains a global problem and more effective methods to prevent CDIs are urgently needed [11,12]. As the knowledge of the roles of the intestinal microbiome expands and its protective abilities are appreciated, newer strategies to prevent CDIs have been explored.

The microbiome of skin and mucosal membranes of humans may consist of up to 100 trillion microbes (bacteria, fungi, viruses, and phages) and 500–1000 different species and functions as an epithelial barrier, immune regulator, and method to resist the colonization of opportunistic pathogens called ‘colonization resistance’, reflecting the many metabolic functions of the microbiome [13,14,15]. The intestinal microbiome is dominated (over 90%) by two phyla: Bacteroidetes and *Firmicutes* (including *Lactobacillus, Bacillus,* and *Clostridioides* ssp.). Recent advances in metagenomic analytic tools (such as 16S rRNA genesequencing, shotgun metagenomic sequences, DNA hybridization, and phylogenetic microarray assays) have allowed a more definitive determination of the types of unculturable microbes present in diverse microbiomes and have allowed insights into which phyla of bacteria and fungus play important roles in this ‘colonization resistance’ and how disease may alter the microbial profiles present in the diseased organ [15,16]. Altered profiles of the intestinal microbiome have been documented as people age and in various diseases, including CDI [17].

The pathogenesis of CDI is closely related to the disruption of the normally protective microbiome in the intestines and exposure to the spores of *C. difficile*. After *C. difficile* spores are ingested and sporulate in the small intestine (triggered by primary bile acids), the vegetative cells reach the large colon [18]. If the normal microbial layer is undisturbed, complex and varied mechanisms prevent *C. difficile* from attaching to intestinal enterocytes, thus preventing disease. However, if the microbiome has been disrupted (typically by antibiotic exposure), this ‘colonization resistance’ is compromised, allowing *C. difficile* cells to attach to enterocytes, reproduce, and release several types of toxins. Toxin A and B, and binary toxins can disrupt the cytoskeleton, opening tight junctions that release fluid (diarrhea), and the dying enterocytes also attract cytokines, causing an inflammatory response, which may result in colitis or the formation of plaques (pseudomembranous colitis). The intestinal tract also provides an immune response (anti-toxin antibodies), and strategies to enhance this immune response have also been investigated [18].

CDI in adults and neonates has resulted in intestinal dysbiosis and a reduction in the diversity of several normal phyla (*Firmicutes*, *Bacteroidetes,* and *Actinobacteria*) and an increase in *Proteobacteria* [17,19]. Studies of patients with CDI have found the loss of these microbes results in a decrease in butyric acids and other short-chain fatty acids, a reduction in bile acids, alterations in nutrient availability, changes in intestinal cell physiology, and immune system changes [13]. Infants and neonates may have high asymptomatic carriage rates of toxigenic and non-toxigenic strains of *C. difficile*, but typically do not show symptoms, perhaps due to lack of mature toxin attachment sites [19]. The composition of the intestinal microbiome of asymptomatic carriers of *C. difficile* was found to have more *Clostridial* sp. capable of degrading simple carbohydrates compared to species found in symptomatic CDI patients [20]. As the replication of *C. difficile* vegetative cells is dependent upon the presence of simple carbohydrates, the difference in the microbiome may explain why asymptomatic carriers may not develop subsequent *C. difficile* disease. Early studies found decreased diversity (less *Bacteroidetes* and *Firmicutes*) in patients with recurrent CDI [21]. More recent studies have confirmed identified differences in the composition of the intestinal microbiome in patients with recurrent CDI, with lower levels of beneficial microbes (*Ruminococcaceae*, *Bacteroidaceae,* and *Lachnospiraceae*) and higher levels of detrimental bacterial species (including *Veillonella*, *Enterobacteriaceae*, and *Streptococcus* sp.) compared to patients without recurrence [17,22].

Our aim in this review is to update our present knowledge of the primary prevention, the initial episode of CDI, and the secondary prevention of recurrences of CDI and to also update the progress of microbiome-targeted investigational strategies.

## 2. Materials and Methods

We reviewed the literature from 2018 to 2023 using databases (PubMed, Google Scholar, clinicaltrials.gov, accessed on 3 March 2023) for recent clinical trials for primary or secondary prevention of CDI and therapies involving interactions with the intestinal microbiome (from 2018–3 March 2023). Search terms included *Clostridioides*, *C. difficile*, recurrent *C. difficile*, prevention, treatment, vaccines, probiotics, fecal microbial therapy, live biotherapeutic products, microbiome, safety, and risks. Inclusion criteria included phase 2 or 3 randomized controlled trials, case-control studies, or quasi-experimental studies for the prevention or treatment of *C. difficile* infections, adults or pediatrics, inpatients or outpatients, and studies on the mechanism of action directed at the intestinal microbiome. If no phase 3 trials were published in the past five years (2018–2023), we included the results from the most recently published phase 2 trial. No language restrictions were in place. Exclusion criteria included studies on other types of diseases and early phase 1 or 2 studies on safety, formulation, or dose-ranging that did not evaluate efficacy outcomes. We also explored the grey literature using queried references cited from reviews and published papers.

## 3. Results

### 3.1. Strategies for Primary Prevention of CDI

Primary CDI prevention aims to reduce the number of people developing their first CDI episode, while secondary prevention aims to reduce the frequency of patients developing one or more recurrent CDI episodes. Developing strategies to prevent CDI requires an understanding of how *C. difficile* is transmitted and which factors may increase the risk of a person developing an initial episode of CDI or recurrent episodes. The transmission of *C. difficile* has been well documented by many studies in healthcare settings since 1989 and show *C. difficile* spores are shed by CDI patients to environmental surfaces, may be carried on hands and clothing of hospital staff and visitors, and may infect roommates and hospital personnel [23,24]. The source of community-acquired cases of CDI may be spores from soil, animals, or asymptomatic carriers recently discharged from healthcare facilities [25,26]. The spores of *C. difficile* are resistant to non-sporicidal disinfectants and may persist in the environment for up to one year after being shed [24]. Risk factors for CDI include demographic factors (age over 65 years, immunocompromised, or prior CDI episodes), exposure to medications or procedures (antibiotics, chemotherapy, intestinal surgery, or enteral feeding), or environmental factors (admission to hospital or long-term care facility, contact with CDI patient, contact with colonized healthcare staff, prior hospitalization, or community sources) [27,28].

Targets for preventive measures involve four main pathways: (1) interrupting environmental transmission of *C. difficile* spores, (2) modifying CDI risk factors, (3) boosting the immune system, and (4) reinforcing the normally protective intestinal microbiome (Figure 1).

#### 3.1.1. Interruption of *C. difficile* Transmission

Enhanced infection control programs using early detection of CDI cases, surveillance, isolation or cohorting of CDI cases, contact precautions, hand hygiene training, sporicidal disinfection of environmental surfaces, and terminal room cleaning have resulted in a decrease in CDI rates in many healthcare settings and are recommended by the Infectious Disease Society Association (IDSA) guidelines [29,30]. Another study initiated an infection control bundle including antibiotic stewardship, patient isolation, early detection of CDI, and increased hand washing education and found CDI rates fell from 0.3% admissions before the infection control bundle was implemented to 0.1% admissions afterward (*p* = 0.035) [31].

#### 3.1.2. Modifying CDI Risk Factors

Modifying CDI risk factors can be more difficult, as age and co-morbidities are not flexible, but medication use can be changed. Surveys indicate that 30–50% of antibiotic use is inappropriately given and the implementation of antibiotic stewardship programs (ASPs) has documented a reduction in CDI rates [1]. A meta-analysis of 11 studies showed a 32% reduction in CDI rates after ASPs were implemented [32]. Kullar et al. reviewed ASP studies and recommended several interventions for improving ASPs including documenting adherence to the program and doses of antibiotics used, identifying a local champion for the program, and having administration buy-in to the program [33]. By reducing the amount of inappropriate antibiotics used, fewer patients will have dysbiosis (disrupted microbiome) when exposed to *C. difficile*, which should result in a lower rate of patients susceptible to CDI.

The use of prophylactic antibiotics has been suggested to prevent primary CDI in high-risk patients (immunocompromised patients or predicted systemic antibiotic use) and has resulted in a reduction in incident CDI in a few studies with vancomycin use [34,35], but the use of fidaxomicin did not significantly reduce CDI rates in one study [36]. Current clinical guidelines do not recommend the use of prophylactic antibiotics to reduce CDI [37].

#### 3.1.3. Enhancing the Immune Response

The intestine is the source of the majority of secretory IgA and other antibodies that can protect against CDI. Enhancing the immune response has been targeted by several types of vaccines comprised of anti-toxin A/B toxoids or proteins that interfere with *C. difficile* binding. “Cdiffense” is a toxoid against toxins A and B developed by Sanofi/Pasteur. During the phase 3 trial, the development of this toxoid was discontinued due to a lack of efficacy. In 9302 enrolled subjects, 0.4% of those patients receiving the toxoid developed CDI compared to a similar number (0.51%) of those in the control group (*p* > 0.05) [38]. Another vaccine (“PF-0642590”), a toxoid against toxin A/B developed by Pfizer, has completed a phase 2 dose study. Of 855 enrolled patients, the most effective dose was 200 micrograms given monthly [39]. In a phase 3 trial (CLOVER) of the PF-0642590 vaccine, the study failed to achieve the main outcome of primary prevention of CDI, but the final results have not been published yet. A chimeric protein “VLA84” developed by Valneva binds to the attachment sites of toxins A and B, thus preventing *C. difficile* attachment and infection. A phase 2 safety study of 140 subjects was completed and 55–91% showed toxin A/B neutralizing antibody levels and the same tolerance in treated and control groups [40]. No phase 3 clinical trials of VLA84 have been published or registered with clinicaltrials.gov. Another preventive measure “PP108” is being developed by GSK, consisting of *Bacillus subtilis* spores, which compete with *C. difficile* toxin A binding sites, and has only been tested in hamster models currently and no clinical trials have been published [41].

#### 3.1.4. Enhancing the Intestinal Microbiome with Probiotics

The most active area of preventive measures involves the restoration or reinforcement of the normally protective intestinal microbiome. Probiotics are defined as “living microbes given in adequate amounts which have a health benefit to the host” [42]. As living entities, probiotics have the advantage of multiple mechanisms of action that may be directed against *C. difficile*. Four main pathways have been documented by various probiotic strains: (1) inhibition of *C. difficile* or spore germination with bacteriocins, attachment site competition, changing intestinal pH, or reduction in primary bile salts; (2) toxin A/B inhibition by quorum sensing quenching or anti-toxin protease or lytic peptide production; (3) restoring intestinal cell physiology by butyrate production; and (4) stimulating the immune system (increased levels of IgA) [13,27,43]. Several types of bacterial and fungal probiotics (*Saccharomyces* (*S*.) *boulardii* CNCM I-745, *Bifidobacterium* (*B*.) *lactis* LAFTI B94, *Lactobacillus* (*L.*) *gasseri* OLL2716, *Lacticaseibacillus* (*Lcb*.) *casei* Shirota, *Lactiplantibacillus* (*Lpb*.) *plantarum* 299v, and others) have been shown to restore the disruption of the microbiome associated with disease or antibiotic use [13,44,45,46]. It is important to recognize that the efficacy and impact on the microbiome are typically specific to the strain of the probiotic and the disease being prevented or treated [47]. Thus, the efficacy should be judged for each probiotic strain or mixture of strains separately, and conclusions should not be made based on pooled different types of probiotics.

In the area of probiotics, only five randomized controlled trials (RCTs) with CDI as the primary outcome have been published (Table 1) [48,49,50,51,52]. Plummer et al. enrolled 138 elderly inpatients given antibiotics at a hospital in the United Kingdom [48]. Patients were randomized to either a two-strain blend (*L. acidophilus* and *B. bifidum*) given at a dose of 2 × 10^10^ colony-forming units (cfu) per day for 20 days or a placebo. CDI developed in 2.9% of the probiotic group and 7.2% in the control, but the difference was not significantly different. Of note, the strains of the two probiotics were not reported. Rafiq et al. randomized 100 adult inpatients given oral or intravenous antibiotics to a three-strained blend (of either *L. acidophilus, L. bulgaricus,* or *B. bifidum*) at a dose of 1.2 × 10^10^/d during the antibiotic course [49]. Significantly fewer patients given the probiotic blend developed CDI (11%, *p* < 0.05) compared to the controls (40%). The strain designations for the three probiotic species were not reported. A small RCT of 42 inpatients given antibiotics at a hospital in Israel was randomized to a four-strained probiotic blend (of either *L. acidophilus*, *L. bulgaricus, B. bifidum,* or *Streptococcus thermophilus*) or placebo, but no significant difference in CDI rates was observed for the probiotic blend compared to the placebo group (14.3% and 4.8%, *p* > 0.05, respectively) [50]. Once again, in this older study, the strain designations were not provided. Miller et al. reported two RCTs using *Lcb. rhamnosus* GG at either a daily dose of 4 × 10^10^ or 1.2 × 10^11^, but no significant differences in CDI rates were noted [51].

There is a lack of RCTs using CDI as a primary outcome for other strains of probiotics, making conclusions difficult. We did not find any trials using probiotics for the primary prevention of CDI with CDI as their primary outcome published in the past five years. In addition, these early trials did not document changes to the intestinal flora, so the degree of microbiome restoration was unknown.

Another method to evaluate probiotic strains for CDI is to utilize a meta-analysis to pool data from RCTs that used probiotics to prevent antibiotic-associated diarrhea as the primary outcome, but also reported CDI as a secondary outcome. In a meta-analysis of 22 RCTs, five sub-groups of different types of probiotics were evaluated for the prevention of CDI, as shown in Table 1 [52]. Four probiotics were found to significantly reduce incident CDI: *S. boulardii* CNCM I-745 (RR = 0.52, 95% C.I. 0.31–0.88, based on 10 RCTs), *Lcb. casei* DN14001 (RR = 0.07, 95% C.I. 0.01–0.55, based on two RCTs), a two-strain blend of *L. acidophilus* and *B. bifidum* (RR = 0.41, 95% C.I. 0.21–0.80, based on three RCTs), and a three-strain blend of *L. acidophilus* CL1285 and *Lcb. casei* LBC80R and *Lcb. rhamnosus* CLR2 (RR = 0.21, 95% C.I. 0.11–0.40, based on four RCTs. *Lcb. rhamnosus* GG did not significantly reduce CDI in adults (based on five RCTs). *S. boulardii* CNCM I-745 has been shown to restore the antibiotic-disrupted microbiome quickly in both animal models and humans and results in an increase in diversity (more *Enterobacteria* and a reduction in *Bacteroides* and *Clostridia* spp.) [46,53,54]. The three-strain blend above has been shown to inhibit *C. difficile* sporulation and reduce Toxin A and B levels [55] A review of five meta-analyses (from 2021 to 2022) reviewing 16–31 RCTs concluded some strains of probiotics significantly prevented CDI but noted not all these meta-analyses accounted for strain-specific efficacy [27].

The efficacy of probiotics has also been tested using study designs other than RCTs (quasi-experimental studies or matched case-control studies) for the primary prevention of CDI. In quasi-experimental studies, the probiotic intervention is given to eligible inpatients (typically any inpatient starting a new antibiotic course) and given for the duration of the hospital stay or antibiotic course. Facility-wide CDI rates are compared for a baseline time period (typically one year before the intervention is started) and the probiotic intervention period. Three different probiotics have been tested in this real-life scenario (Table 2) with promising results for all three probiotic types [56,57,58,59,60,61,62,63,64,65]. Three of five studies giving a three-strained probiotic blend (*L. acidophilus* CL1285, *Lcb casei* LBC80R, and *Lcb. rhamnosus* CLR2) to eligible patients were able to significantly reduce facility-wide CDI rates [56,57,58]. For example, at one hospital in Quebec, Canada, this three-strain *Lactobacilli* blend was offered to any eligible patient (admitted adult patient with estimated new antibiotic use for at least two days) using a pharmacy-driven electronic trigger whenever an antibiotic was ordered. The three-strain probiotic was begun within 24 h of the antibiotic and given for the duration of the antibiotic. The incidence of hospital-onset CDI rates was significantly reduced from the baseline period (one year before the program had begun) compared to one year of probiotic administration (8.6/10,000 to 5.2/10,000 patient-days, respectively, *p* = 0.002), and a low CDI rate was observed as the probiotic use was continued for the next 18 months (3.05/10,000) [66]. Lower CDI rates were maintained over the next 15 years using this three-strained probiotic blend [67]. Of the three facilities that tested *S. boulardii* CNCM I-745 on patients receiving antibiotics, only one study showed a significant reduction in CDI rates [63]. Both studies providing patients with another three-strain probiotic blend of *L. acidophilus*, *B. longum,* and *B. bifidum* Bb12 significantly reduced facility-wide CDI rates [64,65].

However, it should be noted that limitations to quasi-experimental studies include (1) CDI rates may be impacted by changes in infection control practices or changes in antibiotic usage during different study periods, (2) a lack of adherence to probiotic administration may differ, (3) a sufficient run-in time for full implementation may be necessary to achieve the best effects, and (4) it may be difficult to detect a significant difference for facilities with a low rate of baseline CDI cases [68].

Currently, recommendations from clinical practice guidelines for the use of probiotics in the primary prevention of CDI vary. The American Gastroenterology Association guidelines recommend four probiotics for the prevention of CDI (*S. boulardii* CNCM 1-745 or a three-strain blend of *L. acidophilus* CL1285 and *Lcb. casei* LBC80R and *Lcb. rhamnosus* CLR2 or a three-strain blend of *L. acidophilus*, *L. bulgaricus,* and *B. bifidum* or a four-strain blend of *L. acidophilus*, *L. bulgaricus*, *B. bifidum,* and *Streptococcus thermophilus* [69]. In contrast, the American College of Gastroenterology guidelines could not conclude any probiotics were effective, but it was later pointed out that these guidelines did not account for strain-specificity [70,71]. The European Society of Clinical Microbiology and Infectious Diseases (ESCMID) guidelines concluded there was insufficient evidence for probiotics for the primary prevention of CDI [37]. This conclusion was based on several factors: (1) a lack of effect seen in a large trial (*n* = 2981 patients) testing a four-strain blend [72], and (2) safety concerns with another study in pancreatitis patients showing higher mortality if given a nine-strain probiotic blend [73]. The guideline did add a positive result was noted in a large (*n* = 8763 patients) study that found a significant reduction in CDI rates if *S. boulardii* CNCM I-745 was administered compared to controls (0.56% and 0.82%, respectively, *p* < 0.05) [63]. It should be noted, however, that the study by Allen et al. was designed for the prevention of antibiotic-associated diarrhea as the primary outcome and not for CDI (only 15% power to detect a significant effect existed) and the higher mortality in the pancreatitis patients was not attributable to the probiotic [72,73].

In studies of probiotics used for the primary prevention of CDI, the use of probiotics has been well tolerated and few serious adverse events have been noted. In general, risks of living microbe administration may involve translocation from the intestinal tract, bacteremia or fungemia, or the transfer of antibiotic-resistant genes [27]. Translocation has been observed in immunosuppressed animal models, but rarely in humans. No cases of transfer of antibiotic-resistant genes have been reported. Bacteremia or fungemia is infrequently reported in immunocompromised or severely ill patients [74]. A review of the literature from 1980 to 2023 found only 23 cases of *Lactobacilli* sp. bacteremia where the blood isolate and the strain of the oral probiotic were identified as identical and most cases (20/23) were caused by *Lcb. rhamnosus* GG [74]. Overall, most probiotic strains have an excellent safety profile and are well tolerated, but caution is recommended if the patient has a severe chronic disease, is immunocompromised, or has a central catheter in place [27].

### 3.2. Strategies for the Secondary Prevention of CDI

Secondary prevention of CDI is aimed at preventing any further episodes after the initial CDI episode resolves. Recurrent CDI (rCDI) is defined as a recurrence of symptoms (typically diarrhea) with a positive *C. difficile* toxin result within eight weeks of the previous CDI episode. Once a patient develops an initial episode of CDI, 16–20% may develop another episode of CDI, and once a second episode of CDI has occurred, 40–60% may develop at least three recurrences [10,75]. Patients with recurrent CDI typically have more severe disease and a higher risk of developing sepsis (17–43%) and are more likely to be re-admitted to a healthcare facility [8].

Non-toxigenic *C. difficile* strains have been tried after the initial CDI episode has been treated with vancomycin or metronidazole in order to interfere with the attachment sites of toxin A or B, thus preventing a recurrence of CDI. One phase 2 dose-ranging study using *C. difficile* VP20621 (NTCD M3) randomized adults with prior CDI to 10^4^/d–10^7^/d for 7 or 14 days or placebo found significantly fewer recurrences in those given the non-toxigenic strain vs. placebo (11% and 30%, respectively, *p* < 0.05), but no phase 3 studies have been published [76].

Standard antibiotics used to treat CDI aim to reduce the severity and duration of symptoms, but also to prevent subsequent recurrence of disease. Current clinical practice guidelines from the Infectious Disease Society of America and Society of Hospital Epidemiologists of America (IDSA/SHEA), ESCMID, and the Australasian Society for Infectious Diseases (ASID) recommend fidaxomicin or vancomycin for non-severe initial CDI episodes or metronidazole if the prior two antibiotics are not available [37,77,78,79]. Recent 2021 guidelines also recommend the use of fidaxomicin (either a standard 10-day or extended course) or a tapered/pulsed regimen of vancomycin over 12 weeks to prevent a second recurrence of CDI [79]. To prevent subsequent CDI (>2 recurrences), bezlotoxumab (an FDA-approved monoclonal antibody against toxin B) or fecal microbial therapy may be added to the antibiotic regimen [37,77,78,79,80]. For severe or complicated CDI cases, nasogastric or rectal administration of vancomycin or surgery may be considered. Even though fidaxomicin is a minimally absorbed macrocyclic antibiotic with only mild impacts on the normal intestinal microbiome, all three antibiotics impact the microbiome to some extent, delaying the recovery of the protective microbial barrier [81]. A recent narrative review described recommendations for primary prevention and treatment of CDI, but did not explore how these therapies interact with the intestinal microbiome [77].

In our review, we provide updates on three investigative strategies for the secondary prevention of CDI that focus on the restoration of the normal protective microbiome: (1) fecal microbial therapy, (2) live biotherapeutic products, and (3) probiotics, as shown in Table 3 [82,83,84,85,86,87,88,89,90].

#### 3.2.1. Fecal Microbial Therapy (FMT)

Fecal microbial therapy (FMT) aims to assist in the restoration of the normal intestinal microbiome that has been disrupted, typically during antibiotic exposure. A fecal solution from carefully screened healthy donors is seeded into the CDI patient’s intestines either via oral capsules, endoscopic delivery, or by enema. Even though the composition of the microbial species varies from donor to donor, successful prevention of CDI recurrences has been reported from 78 to 100% in one meta-analysis of 15 trials [91]. A meta-analysis of 14 studies (including RCTs, uncontrolled case reports, and case series) documented a pooled 86% cure (no CDI recurrences) with low adverse reactions (15%) and only 2% with serious adverse events [92]. Meta-analyses of FMT often pool results from phase 2 safety/formulation studies and phase 3 controlled trials, and CDI cure rates differ depending upon the study design (blinded controlled trials average 67.7% cure vs. 82.7% cure for open uncontrolled studies) [93]. As a consequence, we present only the results of three phase 3 RCTs with FMT (Table 3), which shows two of the three RCTs showed a significant reduction in recurrences of CDI [82,83,84]. No phase 3 RCTs with FMT published since 2017 were found.

One study in nine patients with recurrent CDI found increased diversity (more *Bacteroidetes* and *Clostridioides* sp.) after FMT was given [94]. Two RCTs reported an increase in microbiome diversity and an increase in *Bacteroidetes* and *Clostridioides* cluster IV and XIVa levels after FMT [82,84]. A recent review showed an increase in richness and diversity of the microbiome after FMT, a restoration of butyrate-producing *Bacteroidetes* and *Roseburia*, an increase in beneficial taxa (*Firmicutes*, *Ruminococcaceae*, *Verrucomicrobiaceae*, etc.), and a decrease in *Proteobacteria* [22].

Limitations of FMT include the heterogeneous composition of individual donor stools, the potential transmission of antimicrobial resistance genes after FMT, and the transmission of pathogens from the donor (including SARS-CoV-2, *Blastocystis* sp., multidrug-resistant *E. coli*, or bacteriophages) [95,96,97,98].

#### 3.2.2. Live Biotherapeutic Products (LBP)

In order to address the issue of donor stool heterogeneity in FMT, the development of known species or purified biologics has been developed. Live Biotherapeutic Products (LBP) are evaluated by the Food and Drug Administration (FDA) and several candidates have been tested in phase 2 and phase 3 RCTs (Table 3) [85,86,87,88]. In these studies, all patients had recurrent CDI, had been treated with standard-of-care antibiotics (vancomycin, fidaxomicin, or metronidazole), and were followed for at least eight weeks for CDI recurrences.

RBX2660 (Rebiotix Inc., Roseville, MN)) is a mixture of strains isolated from healthy donor stools and was approved by the FDA on 30 November 2022 for the prevention of CDI recurrences [99]. RBX2660 is given by enema in a single dose (≥10^7^ live microbes per mL) including *Bacteroidetes* and *Firmicutes* sp. among other non-specified species. Each dose is manufactured with quality controls in accordance with FDA LBP guidelines. Each dose is from a single donor who has undergone extensive screening for 29 intestinal pathogens. The safety and efficacy of RBX2660 have been tested in two phase 2 studies [100,101], and in a phase 3 efficacy trial (PUNCH CD3) [85]. Safety data across most of the RBX2660 studies (>1000 patients) showed this LBP was well tolerated with only mild–moderate intestinal adverse events [99]. However, in the PUNCH3 trial, significantly more adverse events were reported in the RBX2660 group (56%) compared to the placebo group (45%), but these were mostly mild–moderate intestinal symptoms. In a phase 3 RCT of 267 patients (PUNCH-CD3 study) with recurrent CDI, significantly fewer patients given RBX2660 developed a subsequent CDI recurrence compared to placebo (29.4% vs. 42.5%, respectively), as shown in Table 3 [85]. The baseline levels of *Bacteroidia* and *Clostridia* sp. were also restored within one week of RBX2660 and increased alpha diversity was documented for at least 6–24 months [102]. Decreased levels of *Gammaproteobacteria* and *Bacilli* species were also noted [101].

Another LBP has been developed by Seres Therapeutics called SER-109, which is a purified preparation of *Firmicutes* spores derived from a healthy donor stool. In a phase 3 RCT of rCDI patients (ECOSPOR III), patients who had symptom resolution after 10–21 days of standard-of-care antibiotics were enrolled and randomized to either four capsules of SER-109 (3 × 10^7^ spores/day) or placebo for three days and followed for 8 weeks for recurrences [86]. Of 182 enrolled patients, significantly fewer patients had another recurrence (12% vs. 40% in placebo, *p* < 0.001). Mild–moderate intestinal adverse events were reported at similar frequencies for the SER-109 group (93%) and 91% of the placebo group. This study also documented a restoration of normal intestinal microbiome profiles after SER-109 was given (increase in *Firmicutes* and decrease in *Enterobacteria* spp.) [103]. An open-label study enrolling 29 patients who failed in the ECOSPOR III trial and an additional 234 patients with rCDI gave SER-109 (four capsules three times a day for 3 days) also found a low rate of recurrence (23, 8.7%) by week 8 [104]. This LBP (named “Vowst” was approved by the FDA on April 26, 2023, for the treatment of recurrent CDI.

CP101 is a mixture of strains isolated from a healthy donor stool (developed by Finch Research). The specific strains have not been reported. In a phase 2 study (PRISM3), 198 patients with recurrent CDI who had completed standard-of-care antibiotics were randomized to either CP101 given as a single oral capsule (6 × 10^11^/capsule) or a placebo capsule and then followed for eight weeks for recurrences of CDI [87]. Patients given CP101 not only had fewer recurrences than the placebo group (26% vs. 41%, respectively, *p* < 0.05), but also showed a higher alpha diversity through week eight [105]. Engraftment of some donor species was found in those given CP101 at eight weeks. The frequency of adverse events was similar for both groups. On long-term assessment (24 weeks), significantly fewer given CP101 had CDI recurrences (27/102, 26.5%, *p* = 0.02) compared to those on placebo (39/96, 40.6%).

Another LBP called VE303 (Vedanta Biosciences Inc.) is an eight-strain mixture derived from a healthy donor stool. The identity of the eight strains has not been revealed. A phase 2 study (CONSORTIUM) enrolled 79 adults with recurrent CDI who were treated with standard-of-care antibiotics. Patients were randomized to capsules of either a lower dose (2 capsules, 2 × 10^10^/d) or a higher dose (10 capsules, 1 × 10^11^/d) or placebo for two weeks and then followed for eight weeks for recurrences [88]. Although no significant difference in CDI recurrence was observed for the lower dose (10/27, 37%, *p* = 0.1), a significant reduction was observed for the higher dose (4/29, 13.8%, *p* = 0.02) compared to the placebo (10/22, 45.5%). Most patients reported mild–moderate adverse events, but the frequency was similar in all three groups. The impact of VE303 on the intestinal microbiome was not reported in this study. No published phase 3 study has been published.

#### 3.2.3. Probiotics

Probiotics have been well studied for the primary prevention of antibiotic-associated diarrhea and CDI, but few phase 3 trials have been conducted to assess the reduction in CDI recurrences. Previous reviews and meta-analyses have reviewed phase 2 and phase 3 trials, but the limited number of trials prevented a strong conclusion of use from being formed [52,69]. Most of the trials were small pilot studies or phase 2 studies that were discontinued early due to poor enrollment or lack of apparent effect [106,107,108,109]. Only two phase 3 RCTs have been conducted assessing the yeast probiotic, *S. boulardii* CNCM I-745 (Table 3) [89,90]. The first study enrolled 124 patients with either initial CDI or recurrent CDI and randomized patients to either *S. boulardii* CNCM I-745 or placebo for 28 days [89]. Overall, significantly fewer recurrences were observed in those patients given *S. boulardii* (26.3%) compared to placebo (44.8%), but the efficacy was limited to those patients enrolled with recurrent CDI (Table 3). The second phase 3 trial enrolled 168 adults with recurrent CDI and combined *S. boulardii* or placebo with one of three antibiotic treatments [high dose (2 g/d) vancomycin (*n* = 32), low dose (500 mg/d) vancomycin (*n* = 83), or metronidazole (1 g/d, *n* = 53)] [90]. A significant reduction in CDI recurrences was only observed for the high-dose vancomycin and *S. boulardii* group. *S. boulardii* has been shown to increase intestinal microbiome diversity and may help to restore the normal microbiome [53]. No further phase 3 RCTs were published for this probiotic and no other phase 3 trials have been published for the other promising probiotics reported in early pilot studies.

## 4. Discussion

As our knowledge about the complex interactions between the host, environmental factors, and the intestinal microbiome expands, newer strategies are being investigated for the primary and secondary prevention of CDI. CDI is an infection that is intimately involved in the dysbiosis of the intestinal microbiome and thus strategies focused on the restoration of this protective microbial layer are paramount.

The most effective methods for primary prevention of the initial CDI episode have involved four areas: interruption of the transmission of *C. difficile* spores in the healthcare setting, modification of risk factors for CDI, development of vaccines, and microbial interventions to help restore the protective microbiome. Our review updated the existing knowledge on these various avenues and we described the progress made in the past five years (2018–2023). Previous studies documented the effectiveness of using infection control bundles consisting of surveillance for CDI cases, enhanced hand washing and use of personal protective equipment, cohorting CDI cases, and sporicidal disinfection of environmental surfaces [23,25]. As the use of antibiotics is the most common cause of microbiome disruption, antibiotic stewardship programs (ASPs) that evaluate rational uses of antibiotics and reduce the inappropriate use of antibiotics have dramatically reduced incident CDI cases. A meta-analysis of 11 studies showed a 32% reduction in CDI rates after ASPs were implemented [32]. Recent efforts have aimed at methods to improve the infection control bundles and improve compliance with ASPs [33]. Improving antibiotic use and reducing the use of proton pump inhibitors have been the only risk factors for CDI that are easily modifiable [28]. Although compliance with these recommended programs has reduced CDI rates, CDIs still are an urgent threat to healthcare facilities [1,29].

The use of vaccines for CDI toxin A/B is another area for the primary prevention of CDI, but the only phase 3 study published so far was discontinued for lack of effect [38], with two other vaccine candidates having just completed phase 2 trials [39,40]. Other vaccine candidates are in early development. The challenge for *C. difficile* vaccines is the choice of a high-risk target population (high-risk antibiotics, advanced age, and/or immunocompromised) and the need to administer the vaccine sufficiently early to obtain protective antibody titers before a triggering event (such as hospital admission or antibiotic administration) [110].

The use of prophylactic antibiotics at admission to a healthcare facility has been effective in reducing subsequent CDI when vancomycin was used [34], but not when fidaxomicin was given [36]. Current ESCMID guidelines do not recommend the use of prophylactic antibiotics [37].

Disrupted taxa of the intestinal microbiome have been reported to increase the risk of CDI [17,19]. A challenge for strategies involving restoring the normally protective microbiome is the wide diversity of investigative interventions and the differing regulatory requirements, which vary by country and type of product [111,112]. Probiotics (living microorganisms when taken in adequate amounts conferring a health benefit for the host) can be available as dietary supplements, approved drugs, or medicinal foods. Other stool-based microbiome products including fecal microbial therapy (suspensions of screened stool from healthy donors) or live biotherapeutic products (more specifically defined strains from healthy donors or purified strains) have different regulatory oversight. LBP require FDA approval before they can be marketed in the USA.

Probiotics have been investigated as a method to restore the disrupted microbiome. Some single-strain and multi-strained blends of probiotics have a long history of use and show evidence-based efficacy for the prevention of antibiotic-associated diarrhea (AAD), of which CDI is part of the AAD spectrum [47,110]. Evidence for efficacy is typically determined based on two phase 3 RCTs with a CDI as the primary outcome. A limitation of the current literature is the scarcity of RCTs that are designed with the primary prevention of CDI as the primary outcome. We only found five RCTs using CDI as a primary outcome and none were published after 2009. Another challenge for RCTs testing probiotics is that it is recommended that probiotic strains be investigated only in settings where the CDI rate is over 5%, as lower rates of CDI would require extremely large study sizes and, in the absence of large CDI outbreaks, this frequency may be difficult to predict [69]. Two other sources of efficacy data were found: meta-analyses pooling RCTs that were performed to test probiotics for the prevention of AAD and facility-level studies. One meta-analysis from AAD trials investigated five different types of probiotics for the primary prevention of CDI and found four types were significantly effective, but only when the data were pooled over multiple RCTs [113]. These meta-analyses found that not all probiotics have efficacy for AAD and CDI and that it is important to consider the efficacy by the strain or by the type of multi-strained blend, as efficacy is strain-specific [47]. The use of facility-level quasi-experimental studies offers a ‘real-life scenario’ design as to how probiotics would be used as routine practice. The advantage is that probiotics can be offered to any admitted patients predicted to require antibiotics during their stay and the probiotic can be started when antibiotics are begun, especially with the use of electronic flags that are triggered when the pharmacy receives a new antibiotic order. The disadvantage of this design is that facility-wide CDI rates are compared (usually for one year before the program and during one year of the implementation) to determine if the probiotic was effective. Thus, any significant changes in antibiotic use or infection control programs can confound the results. Nevertheless, of 10 facility-level studies conducted (five in the past five years), three types of probiotics were able to show a significant reduction in CDI rates and long-term follow-up studies found the CDI reduction could be maintained (for 2–15 years), as long as the probiotic was continued to be used [30,67].

We found six strategies for the secondary prevention of CDI recurrences: (1) standard-of-care (SoC) antibiotics, (2) monoclonal antibodies, (3) non-toxigenic *C. difficile* strains, (4) FMT, (5) LBP, and (6) probiotics. Recommendations for which strategy is based on the severity of CDI episode and the number of prior CDI episodes [78] Typically, for mild–moderate initial CDI episodes, a 10-day course of either fidaxomicin or vancomycin not only speeds the resolution of CDI symptoms, but results in lower rates of CDI recurrence. For more severe cases of CDI or for repeated CDI recurrence, different regimens of SoC antibiotics (extended or pulsed/tapered) or the use of the FDA-approved Bezlotoxumab monoclonal antibody may be considered [10,37]. No recent trials were found for the use of non-toxigenic *C. difficile* strains. The last three strategies rely heavily on the ability to restore the normal microbiome.

Although early trials showed significant efficacy of FMT (averaging an 86% cure rate with 15% adverse events) for the recurrence of CDI, the procedure was unappealing to many patients and the risk of unidentified pathogens from the donor stool raised concerns with this procedure [92]. Another limitation is the heterogeneity of the microbial composition when different donors are used. No recent (published after 2018) phase 3 RCTs using FMT were found. Efforts to improve this strategy included using more specific known strains derived from donors and using oral capsules as a delivery method rather than an enema or colonoscopic delivery. For example, RBX2660 is the first FDA-approved LBP and was given by enema. A new formulation called RBX7455 is a lyophilized form of RBX2660 that is given as an oral capsule. An early phase 1 trial using capsules found even lower CDI recurrence rates (0–20% depending on dose) compared to the enema delivery of RBX2660 [85,114]. Another LBP (SER-109) given as capsules was recently approved by the FDA. Two other BLA completed phase 2 studies showing good tolerance, but no phase 3 trials were found. Although these LBP preparations may be more homogeneous than FMT, a limitation is that the specific microbial strains used are not reported. Continuing research on LBP seems warranted. Specific probiotics have the advantage of having known strain(s) in the product and do not have the risk of pathogen transmission. Some types of probiotics (*S. boulardii* CNCM I-745 and the three-strain *Lactobacilli* blend (Bio-K+)) documented a significant reduction in CDI recurrences [113], but no phase 3 RCTs published within the past five years were found. Although the use of specific probiotic strains shows promise, more RCTs are needed. Different clinical practice guidelines are in general agreement for the use of SoC antibiotics and FMT for recurrences, but they may differ on the recommendations for probiotics. The American Gastroenterology Association (AGA) recommended a single-strain probiotic (*S. boulardii* CNCM I-745, based on nine trials) and three different multi-strain blends (only the three-strain blend of *L. acidophilus* CL1285, *Lbc. casei* LBC80R, and *Lbc. rhamnosus* CLR2 had multiple RCTs) for the prevention of CDI recurrence [69]. In contrast, the American College of Gastroenterology (ACG) recommends against the use of probiotics [70]. A comparison of these two guidelines suggested the lack of accounting for strain-specific efficacy in the ACG guidelines may have confounded their results [71]. Other European guidelines [37,115] also do not recommend the use of probiotics for the primary prevention of CDI due to several factors: the lack of effect seen with a large trial published in 2013 [72], higher mortality seen in a study of pancreatitis patients [73], and a study showing a delay in microbiome recovery [116]. However, the lack of efficacy found in the study by Allen et al. is applicable only to the four-strain probiotic and should not be generalized to all types of probiotics, and, in addition, this study was powered for AAD and only had a 15% power to detect any efficacy for CDI [72], and a more recent trial by Wombwell et al. of 8763 patients did find significant efficacy when another type of probiotic strain was used [63]. The higher mortality in the trial by Besselink et al. in the nine-strain probiotic blend group was not associated with the probiotic, rather was due to more severe disease in this group [73]. The delay in microbiome recovery was only reported for one 11-strain probiotic in a mouse model [116], and thus should not be generalized to other types of probiotics or when patients take probiotics. A review of five meta-analyses (with 16–31 trials/meta-analysis) concluded there is evidence for the efficacy of probiotics, but strain-specificity must be accounted for [27]. Clinical practice guidelines also suggest more RCTs are needed for both the primary and secondary prevention of CDI using these innovative strategies [69,78,79]. The recent IDSA/SHEA guidelines for the management of CDI only discussed antibiotic treatments or FMT and did not review probiotics [78].

Generally, the use of these strategies has been well tolerated. FMT may have the risk of pathogen transmission from the donor stool and the transmission of SARs-CoV-2, but the transmission of other pathogens has been infrequently reported. Probiotics are generally well tolerated, with mild intestinal adverse reactions reported with some strains. The concern of bacteremia or fungemia has led to the contraindication of use in immunocompromised patients or severely ill patients with central catheters in intensive care units, but the rates of these complications are low. A review of Lactobacilli bacteremia only found 23 cases when the *Lactobacilli* blood isolate was identical to the oral *Lactobacilli* probiotic strain [74]. Another review of studies published between 2008 and 2020 found only 14 cases of fungemia associated with *S. boulardii* CNCM I-745 [117].

## 5. Conclusions

Promising approaches for the primary prevention of CDI that involve the restoration of the intestinal microbiome include limiting unnecessary antibiotic use or the use of specific probiotics when antibiotic use is warranted. For the secondary prevention of CDI, three avenues are promising, including the use of fecal microbial therapy, living biological agents, and some probiotic strains. As the main factor for *Clostridioides difficile* infections is the disruption of the normally protective intestinal microbiome, strategies aimed at restoring the microbiome seem most rational; although, more large randomized controlled trials are needed that document the shifts in the microbiome population.

## Figures and Tables

**Figure 1 microorganisms-11-01534-f001:**
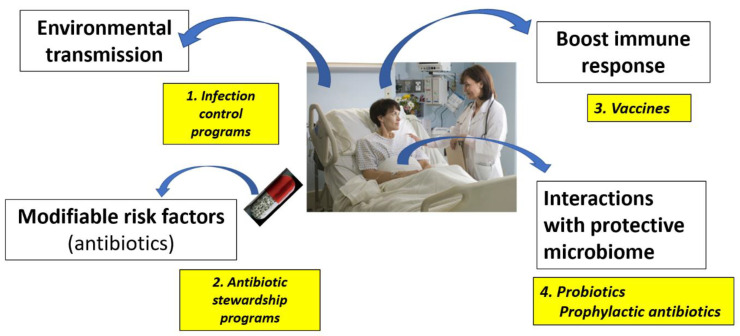
Targets and strategies for the primary prevention of *Clostridioides difficile* infections.

**Table 1 microorganisms-11-01534-t001:** Probiotics tested in most recent randomized controlled trials and meta-analyses for the primary prevention of *Clostridioides difficile* infections.

Study Population	Probiotic Intervention	CDI in Probiotic vs. Control	Reference
Randomized controlled trials			
*N* = 138 elderly inpatients given antibiotics U.K.	2 strain mix (*L. acido.* + *B. bifidum*) 2 × 10^10^ for 20 days	2.9% vs. 7.2% ns	Plummer S 2004 [48]
*N* = 100 inpatient adults or oral/IV antibiotics USA	3 strain mix *(L. acido.*+ *L. bulgaricus* + *B. bifidum*) 1.2 × 10^10^ for duration antibiotics	11% * vs. 40%	Rafiq K 2007 [49]
*N* = 42 inpatient adults on antibiotics Israel	4 strain mix (*L. acido*. + *L. bulgaricus* + *B. bifidum* + *Strept. thermophilus*) 6 × 10^9^ for 3 w	14.3% vs. 4.8% ns	Stein GY 2007 [50]
*N* = 189 inpatient adults on antibiotics USA	*Lcb. rhamnosus* GG 4 × 10^10^ for 2 w	4.2% vs. 7.4% ns	Miller M 2008 [51]
*N* = 316 inpatient adults on antibiotics USA	*Lcb. rhamnosus* GG 1.2 × 10^11^ for 2 w	1.3% vs. 0% ns	Miller M 2008 [51]
Meta-analysis			
*N* = 22 RCTs inpatient adults on antibiotics, separate subgroups by strain(s)	*S. boulardii* CNCM I-745	RR = 0.52 (CI 0.31, 0.88) *	McFarland LV 2017 [52]
	*Lcb. casei* DN114001	RR = 0.07 (CI 0.01, 0.55) *	
	*L. acido. + B. bifidum*	RR = 0.41 (CI 0.21, 0.80) *	
	*L. acido. + Lcb. casei LBC80R +* *Lcb. rhamnosus CLR2*	RR = 0.21 (CI 0.11, 0.40) *	
	*Lcb. rhamnosus GG*	RR = 0.56 (CI 0.29, 1.06) ns	

* *p* < 0.05. Abbreviations: *B*., *Bifidobacterium*; C.I., confidence interval; *L*., *Lactobacillus*; *L. acido.*, *L. acidophilus*; *Lcb*., *Lacticaseibacillus*; *N,* number; ns, not significant; pd, person-days; RCT, randomized controlled trial; U.K., United Kingdom; USA, United States of America; RR, relative risk; *S*., *Saccharomyces*; *Strept*., *Streptococcus;* w, weeks.

**Table 2 microorganisms-11-01534-t002:** Probiotics tested in quasi-experimental studies for the primary prevention of *Clostridioides difficile* infections.

Probiotic	Dose (cfu/d)	# Inpatients on Antibiotics	HO-CDI (during Probiotic vs. during Baseline/Control)	References
*L. acidophilus* CL1285 + *Lbc. casei* LBC80R + *Lbc. rhamnosus* CLR2	5–6 × 10^10^	6548	5.2 vs. 8.6/10,000 pd *	Maziade PJ 2013 [56]
	1 × 10^11^	985	5.5 vs. 6.9/10,000 pd *,**	Trick WE 2018 [57]
	1 × 10^11^	8763	2.8 vs. 7.6/10,000 pd *	Olson B 2015 [58]
	1 × 10^11^	1576	1.7 vs. 0.9/100 ns	Box MJ 2018 [59]
	1 × 10^11^	3291	6 vs. 7.5/10,000 pd ns	Shihadeh K 2018 [60]
*S. boulardii* CNCM I-745	1 × 10^10^	358	9.9 vs. 10.4/10,000 pd ns	Flatley EA 2015 [61]
	2 × 10^10^	not reported	9 vs. 10/10,000 pd ns	Slain D 2020 [62]
	2 × 10^10^	8594	0.6 vs. 0.82/100 *	Wombwell 2021 [63]
*L. acidophilus + B. longum + B. bifidum* Bb12	3 × 10^10^	743	5.5 vs. 16.8/100 *	Graul T 2009 [64]
	3 × 10^10^	43,206	3.9 vs. 4.9/10,000 pd *	Lewis PO 2017 [65]

* *p* < 0.05, ** during 6–12-month intervention. Abbreviations: #, number; *B*., *Bifidobacterium*; cfu/d, colony-forming units per day; HO-CDI, healthcare onset *Clostridioides difficile* infections; *L*., *Lactobacillus*; *Lbc*., *Lacticaseibacillus*; ns, not significant; *S*, *Saccharomyces.*

**Table 3 microorganisms-11-01534-t003:** Most recent randomized controlled phase 3 trials of investigative strategies for the secondary prevention of *C. difficile* recurrences.

Intervention + SoC Antibiotics	Study Population	Dose and Route	Follow-Up (w)	CDI Recurrence in Test vs. Control Group	Reference
FMT	rCDI (*N* = 43)	1X, NG tube	10	19% * vs. 69%	Van Nood E 2013 [82]
FMT	rCDI (*N* = 39)	1–4X colonoscopy	10	10% * vs. 74%	Cammarota G 2015 [83]
FMT	rCDI (*N* = 38)	1X enema	17	56% vs. 42% ns	Hota SS 2017 [84]
RBX2660	rCDI (*N* = 267)	1.5 × 10^9^ 1X enema	8	29.4% * vs. 42.5%	Khanna S 2022 [85]
SER-109	rCDI (*N* = 182)	3 × 10^7^ for 7 d oral capsule	8	12% * vs. 40%	Feuerstadt P 2022 [86]
CP101	rCDI (*N* = 198)	6 × 10^11^ 1X oral capsule	24	26% * vs. 41%	Allegretti JR 2021 [87]
VE303	rCDI (*N* = 79)	2 × 10^10^ 2 w 1 × 10^11^ 2 w oral capsules	8	37% vs. 46% ns 14% * vs. 46%	Louie T 2023 [88]
*S. boulardii* CNCM I-745	iCDI (*N* = 64) rCDI (*N* = 60)	3 × 10^10^ 4 w oral capsules	4	19.3% vs. 24.2% ns 34.6% * vs. 64.7%	McFarland L 1994 [89]
*S. boulardii* CNCM I-745	rCDI (*N* = 32)	3 × 10^10^ 4 w oral capsules	4	16.7% * vs. 50% **	Surawicz C 2000 [90]

* *p* < 0.05 compared to controls, ** high dose vancomycin (2 g/d) subgroup. Abbreviations: CDI, *Clostridioides difficile* infection; cfu/d, colony-forming unit per day; FMT, fecal microbial therapy; f/up, weeks of follow-up post-intervention; iCDI, initial episode of CDI; *N*, number; NG, nasogastric tube; ns, not significant; rCDI, recurrent CDI; SoC, standard-of-care antibiotics (fidaxomicin, vancomycin, or metronidazole); vs., versus; X, number of times given.

## Data Availability

Not available.

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
