# Peer review of "Microbiome-Related and Infection Control Approaches to Primary and Secondary Prevention of Clostridioides difficile Infections"

_microorganisms, 2023, doi:10.3390/microorganisms11061534_

Round 1

Reviewer 1 Report

The manuscript submitted by McFarland et al. is a review on the different approaches for the primary and secondary prevention of C. difficile infection. The authors mainly focussed on the preventive strategies based on microbiome-based therapeutics recently published in the literature (2018-2023). This topic is currently extensively studied and the FDA recently approved the use of specific bacteria or consortium of bacteria to treat recurrent CDI.

The manuscript is very well structured, nicely presented, and easy to read.

To better clarify the concept, could the authors add a paragraph on the definition, differences and regulatory issues between a “traditional probiotic” ,   “Live bio-Therapeutic product” or a “stool-based microbiome therapy” in the introduction or discussion section?

I do not have major comments, only some minor suggestions:

-          Title: “metabiomic” is not frequently used and does not give any results when you launch a search in Pubmed with this word. I suggest to replace this term by “microbiome-based therapeutic” or “biotherapeutic  approach” or even a broader term since the manuscript also covers the infection control measures in the primary prevention of C. difficile

-           Replace “Clostridium” by “Clostridioides” all along the text

-          Introduction: I would also suggest to add a small paragraph on the role of C. difficile immunity since the authors describes the vaccination approach in the primary prevention of C. difficile infections (section 3.1.3)

-          Section 3.1.1: the study to document the efficacy of infection control measure to interrupt CD transmission and decrease CD rates (Olson et al. ) does not seem completely relevant because 1) the reference is just an abstract and not a peer reviewed article, and 2) in this study the infection control bundle also includes the use of two probiotics. I would suggest to replace this reference by another one or to delete it.

-          Line 155: replace “suggestions” by “interventions”

-          Section 3.1.3: the results of the phase 3 of the Pfizer vaccines have been presented at the ECCMID (O0411) and a press release summarizing the major findings is available online (https://www.pfizer.com/news/press-release/press-release-detail/phase-3-clover-trial-pfizers-investigational-clostridioides). Although results were not published yet, it seems relevant to mention that the Phase 3 CLOVER study did not reach its primary endpoint. Regarding VLA84 vaccine candidate, the authors state that the Phase 3 has not been published. I would rather say that no phase 3 is currently recorded in clinicaltrials.gov.

-          The abbreviation of “Bifidobacterium” is “B.” (not Bif.) and that of “Lactobacillus is “L.” (not “Lac.”)

-          Table 2 could be improved by a better separation of the different probiotics

-          Line 291: could the authors add the position of the IDSA (Infection Disease Society of America) regarding their recommendations towards the use of probiotics for C. difficile prevention?

-          Line 420: on April 2023, SER-109 has been FDA-approved to treat CDI recurrence (https://www.contagionlive.com/view/ser-109-is-now-fda-approved-to-treat-recurrent-c-difficile-infection). Please update.

-          Line 449: please update the reference: “Louie T., VE303, a Defined Bacterial Consortium, for Prevention of Recurrent Clostridioides difficile Infection: A Randomized Clinical Trial. JAMA. 2023 Apr 25;329(16):1356-1366. doi: 10.1001/jama.2023.4314.”

-          Line 500: Pfizer has completed the phase 3 clinical trial but the published results are pending.

Reviewer 2 Report

In this review, the authors update the present knowledge on the preventive strategies against Clostridioides difficile infections (CDI) by focusing on papers published in the past five years (2018-2023). They followed literature search and used databases (PubMed, Google Scholar, clinicaltrials.gov) for phase 2-3 clinical trials for primary or secondary prevention of CDI and microbiome and probiotics. Authors point out that strategies aimed at restoring the microbiome seem most rational as the main factor for C. difficile infections is the disruption of the normally protective intestinal microbiome. They highlight some strains of probiotics, use of fecal microbial therapy and living biotherapeutic agents as promising strategies to fill this niche, although spite of need of more large randomized controlled trials that document the shifts in the microbiome population.

However, for publication purpose there are some minor issues the authors should address:

- Line 19. Please use the correct taxonomic name of Clostridioides difficile.

- Paragraph 74-79. It would be interesting to briefly mention the situation of infants who are "protected" against CDI.

- Lines 123. Typo “immune-compromised”.

- Lines 136. Please use italic character for taxonomic names (C. difficile).

- Table 1. The layout of the table, especially the rows in the "Probiotics" column, is not at all clear. Rows in this column are offset from corresponding rows.

- Table 2. The layout of the table is not clear. Rows in this column are offset from corresponding rows. Better separate the 3 probiotics.

- Lines 342. Typo “… of America”.

- Table 3. The layout of the table, especially the rows in the "Dose/duration given (f/up)" column, is not clear. Rows in this column are offset from corresponding rows. Better separate the “Intervention + SoC antibiotics.

- Lines 542. Please add (6 for “probiotics”).

- Lines 612. Please use the correct taxonomic name of Clostridioides difficile.
